# Effect of CaO Sourced from CaCO_3_ or CaSO_4_ on Phase Formation and Mineral Composition of Iron-Rich Calcium Sulfoaluminate Clinker

**DOI:** 10.3390/ma16020643

**Published:** 2023-01-09

**Authors:** Wen Jiang, Changliang Wu, Chao Zhang, Xujiang Wang, Yuzhong Li, Shuang Wu, Yonggang Yao, Jingwei Li, Wenlong Wang

**Affiliations:** 1National Engineering Laboratory for Reducing Emissions from Coal Combustion, Engineering Research Center of Environmental Thermal Technology, Ministry of Education, Shandong Key Laboratory of Energy Carbon Reduction and Resource Utilization, School of Energy and Power Engineering, Shandong University, Jinan 250061, China; 2School of Mechanical Engineering, Tianjin University of Commerce, Tianjin 300134, China

**Keywords:** phase formation, mineral composition, iron-bearing ye’elimite, incorporation

## Abstract

The performance of iron-rich calcium sulfoaluminate (IR-CSA) cement is greatly affected by mineral composition and mineral activity in the clinker. This study aims to identify the effect of CaO sources, either CaCO_3_ or CaSO_4_, on the phase formation and mineral composition of the IR-CSA clinker. Targeted samples were prepared with different proportions of CaCO_3_ and CaSO_4_ as CaO sources at 1300 °C for 45 min. Multiple methods were used to identify the mineralogical conditions. The results indicate that the mineral composition and performance of the IR-CSA clinker could be optimized by adjusting the CaO source. Both Al_2_O_3_ and Fe_2_O_3_ tend to incorporate into C_4_A_3−*x*_F*_x_*$\bar{S}$ with an increase in CaSO_4_ as a CaO source, which leads to an increased content of C_4_A_3−*x*_F*_x_*$\bar{S}$ but a decreased ferrite phase. In addition, clinkers prepared with CaSO_4_ as a CaO source showed much higher *x* value in C_4_A_3−*x*_F*_x_*$\bar{S}$ and higher compressive strength than clinker prepared with CaCO_3_ as the sole CaO source. The crystal types of both C_4_A_3−*x*_F*_x_*$\bar{S}$ and C_2_S were also affected, but showed different trends with the transition of the CaO source. The findings provide a possible method to produce IR-CSA cement at a low cost through cooperative utilization of waste gypsum and iron-bearing industrial solid wastes.

## 1. Introduction

Calcium sulfoaluminate (CSA) cement, which was first developed in China 40 years ago, has attracted increasing attention due to its lower energy consumption and CO_2_ emissions than Portland cement (PC) [1,2,3]. CSA clinker consists mainly of calcium sulfoaluminate (C_4_A_3_$\bar{S}$), dicalcium silicate (C_2_S), and ferrite phases (C_2_F~C_6_A_2_F). It can be produced at 1200 °C–1300 °C, which is about 150 °C lower than PC clinker production [4]. CSA cement exhibits excellent performance in engineering construction, such as precast products, 3D printing, marine-engineering concrete and cold environments, due to its characteristics of high early-age strength, rapid setting, low permeability, and low alkalinity [5,6,7,8]. Therefore, CSA cement will be a promising alternative to PC in future.

The high cost of raw meal has limited the massive scale application of CSA cement in engineering construction, especially expensive bauxite. To reduce production costs, CSA has been produced by using industrial solid wastes (ISWs) containing aluminum, such as aluminum ash and red mud, instead of high-grade bauxite [4,7,8,9]. However, aluminum-bearing ISWs such as red mud, steel slag, and tailings are rich in iron. The presence of iron has an impact on the formation, composition, and hydration activity of the key mineral phases in CSA clinker [10,11,12,13]. Iron is reported mainly to transform to the ferrite phase and ye’elimite [14] and that will make difference to the cement performance. The ferrite phase is beneficial to the early strength, but the ye’elimite is beneficial to both the early and late compressive strength [15,16]. More importantly, the incorporation of iron into ye’elimite makes it possible to decrease the aluminum content in raw materials and expand the aluminum sources. Therefore, researchers have tried to introduce more Fe_2_O_3_ into C_4_A_3_*_−_*_x_F_x_$\bar{S}$ in iron-rich calcium sulfoaluminate (IR-CSA) clinker.

Several researchers produced IR-CSA clinker in CaO-SiO_2_-Al_2_O_3_-Fe_2_O_3_-SO_3_-based systems and found that the *x* value in C_4_A_3−*x*_F_x_$\bar{S}$ is greatly affected by the Fe_2_O_3_, SO_3_, and CaO proportions in the raw mixture [14,17,18,19]. Studies have confirmed that the *x* value in C_4_A_3−*x*_F*_x_*$\bar{S}$ presents a rising trend with the increase in Fe_2_O_3_ content in the raw material; however, not all Fe_2_O_3_ can be substituted for Al_2_O_3_ to participate in the formation of C_4_A_3−*x*_F*_x_*$\bar{S}$ [18]. Low CaO and rich SO_3_ content in raw mixtures have also proven effective in promoting the incorporation of Fe_2_O_3_ into C_4_A_3−*x*_F*_x_*$\bar{S}$ in calcium sulpholuminate ferrite–based systems [17,19]. It is obvious that both CaO and SO_3_ could be provided by CaSO_4_. Thus, it is feasible to ensure sufficient SO_3_ by batching excessive CaSO_4_ as CaO sources in the raw material.

CaO is derived from both CaCO_3_ and CaSO_4_ during the sintering process. Some researchers have indicated that the content of CaSO_4_ as a CaO source has an influence on the mineral composition of the clinker and further affects the performance of the cement. They have reported that a small amount of CaSO_4_ as a CaO source makes no obvious difference to the mineral composition of the clinker and that the loss of SO_3_ was compensated to promote adequate formation of ye’elimite; however, excess CaSO_4_ as a CaO source was reported to lead to the formation of gehlenite, thereby decreasing the compressive strength of the cement [20,21]. Others have indicated that CSA clinker could be obtained with CaO as provided by the partial decomposition of CaSO_4_, with both the key mineral phase composition of the clinker and the properties of the cement being similar to that produced by traditional methods (CaO source from CaCO_3_) [22]. In particular, CSA clinker with CaSO_4_ as the entire CaO source was produced with adequate formation of key mineral phases [23]. In our previous study, we found that batching the slight excess of CaSO_4_ in the raw mixture is an effective method of promoting the utilization rate of Al_2_O_3_ (to form C_4_A_3−x_F_x_$\bar{S}$) [17,20]. In addition, the crystal structure of ye’elimite transformed from orthorhombic symmetry to cubic symmetry because of the incorporation of Fe_2_O_3_, and the x value of C_4_A_3−x_F_x_$\bar{S}$ increased with the increase in CaSO_4_ content. We also prepared IR-CSA with gypsum as the entire CaO source and found that the added iron did not form the ferrite phase first but was incorporated into other phases such as ye’elimite [14]. From the above studies, we found that the phase formation, transformation, and composition of IR-CSA clinker is greatly affected by CaO sources. However, the conclusions of the above studies are slightly biased due to the different preparation conditions. Systematic research into the effect of CaO sources on the phase formation and mineralogic conditions of IR-CSA clinker are still limited, and the mechanism remains unclear.
(1)$$Ⅰ\;3\mathrm{CaO}+3{\mathrm{Al}}_{2}O_{3}+{\mathrm{CaSO}}_{4}\overset{900\sim 950\;℃}{\to }\;3\mathrm{CaO}\cdot 3{\mathrm{Al}}_{2}O_{3}\cdot {\mathrm{CaSO}}_{4}$$
(2)$$Ⅱ\;\mathrm{CaO}+{\mathrm{Al}}_{2}O_{3}\overset{1050\sim 1150\;℃}{\to }\;\mathrm{CaO}\cdot {\mathrm{Al}}_{2}O_{3}$$
(3)$$3\left(\mathrm{CaO}\cdot {\mathrm{Al}}_{2}O_{3}\right)+{\mathrm{CaSO}}_{4}\overset{1050\sim 1150\;℃}{\to }\;3\mathrm{CaO}\cdot 3{\mathrm{Al}}_{2}O_{3}\cdot {\mathrm{CaSO}}_{4}$$
(4)$$Ⅲ\;2\mathrm{CaO}+{\mathrm{Al}}_{2}O_{3}+{\;\mathrm{SiO}}_{2}\overset{850\sim 900\;℃}{\to }\;2\mathrm{CaO}\cdot {\mathrm{Al}}_{2}O_{3}\cdot {\mathrm{SiO}}_{2}$$
(5)$$3\mathrm{CaO}+3\left(2\mathrm{CaO}\cdot {\mathrm{Al}}_{2}O_{3}\cdot {\mathrm{SiO}}_{2}\right)+3{\mathrm{CaSO}}_{4}\overset{1000\sim 1100\;℃}{\to }\;3\mathrm{CaO}\cdot 3{\mathrm{Al}}_{2}O_{3}\cdot {\mathrm{CaSO}}_{4}+3\left(2\mathrm{CaO}\cdot {\mathrm{SiO}}_{2}\right)$$
(6)$$2\mathrm{CaO}+{\mathrm{SiO}}_{2}\overset{1000\sim 1100\;℃}{\to }\;2\mathrm{CaO}\cdot {\mathrm{SiO}}_{2}$$
(7)$$2\mathrm{CaO}+{\mathrm{Fe}}_{2}O_{3}\overset{1000\sim 1100\;℃}{\to }\;2\mathrm{CaO}\cdot {\mathrm{Fe}}_{2}O_{3}$$
(8)$$2\left(2\mathrm{CaO}\cdot {\mathrm{Fe}}_{2}O_{3}\right)+\mathrm{CaO}+\mathrm{CaO}\cdot {\mathrm{Al}}_{2}O_{3}\overset{1150\sim 1200\;℃}{\to }\;6\mathrm{CaO}\cdot {\mathrm{Al}}_{2}O_{3}\cdot 2{\mathrm{Fe}}_{2}O_{3}$$
(9)$$2(2\mathrm{CaO}\cdot {\mathrm{SiO}}_{2})+{\;\mathrm{CaSO}}_{4}\overset{1150\sim 1200\;℃}{\to }4\mathrm{CaO}\cdot 2{\mathrm{SiO}}_{2}\cdot {\mathrm{CaSO}}_{4}$$
(10)$$\;4\mathrm{CaO}\cdot 2{\mathrm{SiO}}_{2}\cdot {\mathrm{CaSO}}_{4}\overset{1200\sim 1250\;℃}{\to }\;2\left(2\mathrm{CaO}\cdot {\mathrm{SiO}}_{2}\right)+{\;\mathrm{CaSO}}_{4}$$
(11)$$6\mathrm{CaO}\cdot {\mathrm{Al}}_{2}O_{3}\cdot 2{\mathrm{Fe}}_{2}O_{3}+\mathrm{CaO}+\mathrm{CaO}\cdot {\mathrm{Al}}_{2}O_{3}\overset{1250\sim 1300\;℃}{\to }\;2\left(4\mathrm{CaO}\cdot {\mathrm{Al}}_{2}O_{3}\cdot {\mathrm{Fe}}_{2}O_{3}\right)$$
(12)$$4\mathrm{CaO}\cdot {\mathrm{Al}}_{2}O_{3}\cdot {\mathrm{Fe}}_{2}O_{3}+\mathrm{CaO}+\mathrm{CaO}\cdot {\mathrm{Al}}_{2}O_{3}\overset{1250\sim 1300\;℃}{\to }\;6\mathrm{CaO}\cdot 2{\mathrm{Al}}_{2}O_{3}\cdot {\mathrm{Fe}}_{2}O_{3}$$

To make clear the effect of the CaO source, as either CaCO_3_/limestone or CaSO_4_/gypsum, on the mineralogic conditions of IR-CSA clinker, the targeted clinker was prepared with the mineral proportion of C_4_A_3_$\bar{S}$:C_2_S:C_4_AF as 50:30:20 by mass. Five groups of raw materials with increasing proportions of CaSO_4_ as a CaO source were designed; namely, whole CaCO_3_ (the CaSO_4_ was assumed to be undecomposed), 90 at.% CaCO_3_ + 10 at.% CaSO_4_ (at.% is the abbreviation form of the atomic ratio in the text), 80 at.% CaCO_3_ + 20 at.% CaSO_4_, 50 at.% CaCO_3_ + 50 at.% CaSO*_4_*, 20 at.% CaCO_3_ + 80 at.% CaSO_4_ and 100 at.% CaSO_4_ were designed to produce the clinkers. To ensure that the mineral formation was complete, all clinker was prepared at 1300 °C for 45 min. The mineralogic conditions as well as the microstructure and the chemical composition of key minerals, which would be influenced by the CaO source, were identified by multiple methods. Finally, the compressive strength was tested as a supplemental validation to verify the feasibility of optimizing the mineralogic conditions of the IR-CSA clinker and further the performance of the IR-CSA cement by adjusting the content of CaSO_4_ as the CaO source.

## 2. Materials and Methods

### 2.1. Raw Mixture

The raw materials (CaCO_3_, Al_2_O_3_, CaSO_4_·2H_2_O, Fe_2_O_3_, and SiO_2_) for preparing the IR clinker were analytical reagents provided by the Sinopharm Group Chemical Reagent Co., Ltd., Shanghai, China. CaCO_3_ was used as the CaO source. CaSO_4_ provided both CaO and SO_3_ during the calcination process of the IR-CSA clinker.

The targeted mineral composition of all clinker and the contents of the raw materials in the preparation of 100 g clinker with chemical reagents are shown in Table 1. The Bogue method was used to obtain the corresponding raw mixture [17]. The clinkers were named as S00, S10, S20, S50, S80, and S100, based on the content of CaO in the form of CaSO_4_. C_m_ indicates the degree of CaO in the raw material in order to satisfy that required for the formation of useful minerals in the corresponding clinker. The formula of C_m_ is defined as in Equation (13), and the value is fixed as 1.00 in this study. The Fe_2_O_3_/(Al_2_O_3_ + Fe_2_O_3_) ratio was the same for all the clinker in this study.
(13)$$C_{m}=\frac{w\left(\mathrm{CaO}\right)-0.7w\left({\mathrm{TiO}}_{2}\right)}{0.73\left[w\left({\mathrm{Al}}_{2}O_{3}\right)-0.64w\left({\mathrm{Fe}}_{2}O_{3}\right)\right]+1.4w\left({\mathrm{Fe}}_{2}O_{3}\right)+1.87w\left({\mathrm{SiO}}_{2}\right)}$$

### 2.2. Preparation Method

The reagents were dried at 110 °C for 2 h in a muffle furnace (LYL-16MA, LUO YANG LIYU KILN CO., LTD, Luoyang, China) and were then ground together in an agate bowl for 1 h to ensure the homogenization of the raw mixture; alcohol was added as a dispersant in the grind progress. After homogenization, 50 g per sample of raw mixture was added to the stainless steel mold (Tianjin Jingtuo Instrument Technology Co., LTD, Tianjin, China); a columnlike raw mixture with a 35 mm diameter was obtained under a pressure of 20 MPa. Thereafter, the obtained columnlike raw mixtureswere fired in an elevator-hearth furnace (LYL-17SJ, LUOYANG LIYU KILN CO., LTD, Luoyang, China) at 1300 ℃ for 45 min and then rapidly cooled at room temperature. Finally, the IR-CSA cement was obtained by milling the clinkers and gypsum. The gypsum added to the clinkers to react with ye’elimite would promote the formation of C_6_A$\bar{S}$_3_H_32_ (AFt) according to Equation (14), but samples with higher M ratios hydrated too fast to form cracks between AFt [24] and thus decreased the mechanical properties of the cement. The optimal gypsum content mixed with clinkers were samples with 0.8 m ratios of gypsum- (the anhydrate in clinkers was involved) to-ye’elimite (M) suggested by [10] and [25]. The content of ye’elimite and anhydrate were determined by the Rietveld refinement method [26]. Subsequently, the cement was prepared as cubic specimens (20 mm × 20 mm × 20 mm) with the water-to-cement ratio of 0.28.
(14)$$C_{4}A_{3}\bar{S}+2C\bar{S}H_{x}+H_{38-2x}\to C_{6}A{\bar{S}}_{3}H_{32}+2{\mathrm{AH}}_{3}-\mathrm{gel}$$
where C_4_A_3_$\bar{S}$, C$\bar{S}$H_x_ and C_6_A$\bar{S}$_3_H_32_ are the abbreviated form of 3CaO Al_2_O_3_ CaSO_4_, CaSO_4_ x H_2_O and 6CaO Al_2_O_3_ SO_3_ 32H_2_O, respectively, and the superscript on S represents sulfate in the compound.

### 2.3. Testing Methods

The phenyl formic acid–ethyl alcohol titration method was adopted to determine the f-CaO content of the IR-CSA clinker according to the Chinese standard GB/T 176–2017.

The sulfate loss was calculated according to the difference in the SO_3_ amount between the raw materials and the clinkers. The SO_3_ contents were determined according to the barium sulfate precipitation in accordance with the Chinese standard GB/T176–2008 [23]. The gypsum decomposition rate is expressed as follows:(15)$$\mathrm{Decomposition}\;\mathrm{rate}\;(\%)=\frac{M_{1}\times S_{1}-M_{2}\times S_{2}}{M_{1\;}\times S_{1}}$$
where *M*_1_ and *S*_1_ represent the mass and SO_3_ content of a raw sample, respectively; *M*_2_ and *S*_2_ are the mass and SO_3_ content of a clinker sample, respectively.

The mineral phase assemblages of all clinkers were determined by an X-ray diffractometer (Aeris, Malvern Panalytical, Malverin, UK) with Cu-Kα radiation. The voltage and the current were set to 40 KV and 15 mA, respectively. The detection range of the X-ray diffraction spectra was set to 10–60° with a scanning speed of 0.0027° per step. The mineral composition was quantified by Rietveld refinement using Topas-Academic V6 software. The crystallographic structures of the phases involved are listed in Table 2. The refined parameters included the background coefficients, instrument parameters, zero-shift error and unit cell parameters [14].

The microstructure of the pattern was acquired by scanning electron microscopy (SEM, Quattro S, Thermo Fisher Scientific, Waltham, MA, USA) with an acceleration voltage of 5 kV.

The chemical composition of clinkers was obtained by a field emission electron probe (FE-EPMA, JXA-8530F-Plus, Tokyo, Japan), and the backscattered electron (BSE) images and elemental distribution images of the patterns were detected with an acceleration voltage of 15 KV and an electron beam current of 5 × 10^−8^ A. The samples were first completely covered with epoxy resin and polished to a mirror state; the prepared patterns were then sprayed with carbon to improve their electrical conductivity for microanalysis.

The cement slurry was fully stirred with a water-to-cement ratio of 0.28 and was then poured into a 20 mm × 20 mm × 20 mm mold. The cement slurry specimens were demolded after curing with a temperature of 20 ± 1 ℃ and 99% relative humidity after 24 h. They were then cured for a standard time of 1, 3 and 28d to test the compressive strength [27].

## 3. Results and Discussion

### 3.1. Decomposition Rate of CaSO_4_ to Free-CaO

The decomposition of CaSO_4_ determines the actual CaO content participating in the reactions. In addition, the decomposition of CaSO_4_ has a great influence on the phase formation, phase composition, and even the chemical composition of the mineral phases according to the study of [14,17]. However, the decomposition of CaSO_4_ to CaO is affected by its proportion in the raw mixture [20]. Thus, it is necessary to confirm the decomposition rate after the calcination. Figure 1 shows the quantitative and theoretical sulfate loss of the clinkers with different CaO sources produced at 1300 ℃. As shown in Figure 1, although the same ye’elimite content was designed in clinkers, the actual SO_3_ content differs in all clinkers. The SO_3_ content increased gradually from clinkers S00 to S80, but then decreased in S100.

The gypsum decomposition rates of all clinker were lower than the theoretical value except for S00. In S00, all CaSO_4_ was assumed to participate in the reactions to form ye’elimite. In fact, there was 14.7 wt.% sulfate loss in the clinker, which led to a higher CaO than the stoichiometric value. However, there was no f-CaO observed in the sample, which means that the extra CaO was involved in the reactions. Extra CaO induces phase equilibrium with a greater ferrite phase (with the CaO/(Al_2_O_3_ + Fe_2_O_3_) ratio of 2.0) formed rather than ye’elimite (with the CaO/(Al_2_O_3_ + Fe_2_O_3_) ratio of 1.3), because of the higher CaO/(Al_2_O_3_ + Fe_2_O_3_) ratio [28]. In clinkers S10−S100, the sulfate loss was less than the designed value and resulted in less CaO participating in the reactions, which led to more Fe_2_O_3_ being incorporated into the ye’elimite [17]. The CaO and SO_3_ content of the clinkers and the differences between the quantitative and theoretical values are listed in Table 3. The differences caused by different proportions of CaCO_3_/CaSO_4_ affected the formation of the mineral phases.

The actual CaO content involved in the reactions is calculated as in Equation (16). CaCO_3_ was assumed to decompose completely above 900 °C and to participate in all reactions. CaO from CaSO_4_ was calculated due to the SO_3_ content.
(16)$$\mathrm{CaO}={\mathrm{CaCO}}_{3}\times \frac{M_{\mathrm{CaO}}}{M_{\mathrm{CaCO}3}}+({\mathrm{CaSO}}_{4}\times \frac{M_{\mathrm{CaO}}}{M_{\mathrm{CaSO}4}}-{\;\mathrm{SO}}_{3}\times \frac{M_{\mathrm{CaSO}4}}{M_{\mathrm{SO}3}}\times \frac{M_{\mathrm{CaO}}}{M_{\mathrm{CaSO}4}})$$

### 3.2. Mineralogical Characterization

#### 3.2.1. Qualitative Phase Analysis

The XRD patterns of the IR-CSA clinker are shown in Figure 2. The key mineral phases, such as ye’elimite and belite existed in all the clinkers, while anhydrite and brownmillerite were partially present. The characteristic peak of anhydrate only presented in clinker S10–S80 and became stronger gradually, which is consistent with the SO_3_ content as tested in Section 3.1. The characteristic peak of brownmillerite (around 12.1°) in S00 is the strongest and becomes weaker gradually with the increase in CaSO_4_ as a CaO source. The characteristic peak of brownmillerite (around 12.1°) vanished in S50 and S80, but its intensity increased slightly in S100. On one hand, more CaSO_4_ batching in the raw material is beneficial for Al_2_O_3_ to form ye’elimite rather than brownmillerite. With the increase in CaSO_4_ as the CaO source, more CaO or CA is wrapped by CaSO_4_ so that the formation paths of ye’elimite expressed as Equations (1)–(3) are prior to that of the ferrite phase expressed as Equations (7)–(11). On the other hand, the residual CaSO_4_ in the clinker means the low CaO content participates in the reactions, and this induces more Fe_2_O_3_ to be incorporated into C_4_A_3-*x*_F*_x_*$\bar{S}$ than to the form ferrite phase [17].

The characteristic peaks of C_4_A_3_$\bar{S}$ and C_2_S were also different between the clinkers. The characteristic peak of C_4_A_3_$\bar{S}$*-o* (at about 18.1°) was obvious in S00 and became weaker, gradually vanishing when CaSO_4_ provided more than 50% CaO. Moreover, the main characteristic peaks of ye’elimite (at approximately 23.7° and 27.4°) gradually shifted to lower angles with the increase in the CaO derived from CaSO_4_ (Figure 3a,b). This was mainly because the interplanar spaces of ye’elimite (with the crystal plane exponent of (022) in C_4_A_3_$\bar{S}$-*o* and (211) in C_4_A_3_$\bar{S}$-*c*, respectively) were increased with the amount of Al^3+^ (0.535 Å) substituted by Fe^3+^(0.645 Å) [17]. Similarly, the crystal structure of belite was additionally affected by CaO sources and the crystal structure of α’-C_2_S was more likely to occur when the CaO source of CaSO_4_ was increased (Figure 3c). The most convenient condition for the formation of α’-C_2_S is that CaSO_4_ provides CaO with the proportion of 50–80 at.% and with anhydrite residue in the clinker.

#### 3.2.2. Quantitative Phase Analysis

Quantitative analysis was conducted using the Rietveld refinement to better understand the formation and transformation of the key mineral phases affected by CaO sources. The results (Figure 4) indicate that the C_4_A_3_$\bar{S}$-o decreased while the C_4_A_3_$\bar{S}$-c increased with the increase in CaSO_4_ as a CaO source, which indicates that more Fe_2_O_3_ is incorporated into C_4_A_3-*x*_F*_x_*$\bar{S}$ to stabilize C_4_A_3_$\bar{S}$ as a pseudocubic crystal during the cooling process [29,30]. The crystal types of belite were also affected by CaO sources. The results indicate that C_2_S only existed as a beta crystal type in S00; β-C_2_S subsequently decreased gradually from S00 to S80, but increased in S100. However, the α’-C_2_S phase showed the opposite trend. The literature has reported that the crystal structure of C_2_S is correlated to its formation paths and that C_2_S as obtained from Equations (5) and (6) exists as β-C_2_S, while from Equation (10) as α’-C_2_S [15]. It is clear that the increasing content of CaSO_4_ induces a greater $4\mathrm{CaO}\cdot 2{\mathrm{SiO}}_{2}\cdot {\mathrm{CaSO}}_{4}$ formation as the transition phase and finally decomposes as α’-C_2_S.

The ferrite phase usually refers to a solid solution with the chemical composition of the formula Ca_2_(Al_y_Fe_2-y_)O_5_, where y can vary from 0 to 4/3. The ferrite phase, with a chemical composition of y = 0, is specified as srebrodolskite (C_2_F), and the set of assemblages with 0 < y < 4/3 usually have a chemical composition approximating to brownmillerite (C_4_AF) [17]. The formation of brownmillerite is considered to comprise a continuous solid solution of calcium aluminate with srebrodolskite (C_2_F) [31]. In this study, both srebrodolskite and brownmillerite existed in all clinker except for S00. There was no srebrodolskite in clinker S00, which shows that srebrodolskite reacts with calcium aluminate to form brownmillerite. In addition, the content of brownmillerite is slightly higher than the designed value; the main reason for this is that the CaSO_4_ is insufficient to react with CA to form C_4_A_3_$\bar{S}$ due to the partial decomposition, and more CA will be a solid soluble in brownmillerite according to Equations (11) and (12). The content of brownmillerite decreases gradually with the increase in CaSO_4._ as the CaO source. In addition, the brownmillerite is much lower than the designed value in the clinker when CaSO_4_ provides the CaO. Small amounts of srebrodolskite were present in clinker S10–S100, which shows that CA is more likely to act with CaSO_4_ than C_2_F when CaSO_4_ batches in raw material as the CaO source. 

### 3.3. Chemical Composition of Key Mineral Phases in Iron-Rich Sulfoaluminate Clinker

The crystal type of ye’elimite is determined by the incorporation level of Fe_2_O_3_, and be’lite is affected by the incorporation of SO_3_ and Fe_2_O_3_, to inhibit crystal transformation during cooling. It is clear that the chemical composition of the mineral phase is affected by the CaO source, in accordance with the results specified in Section 3.2, and this could be determined by the elemental distribution obtained from the polished sections of the sample slices by EPMA; the results are shown in Figure 5 (back-scattered electron (BSE) micrograph Figure 5a—S00, Figure 5b—S50, Figure 5c–S80, Figure 5d—S100) and Figure 6 (a: with the back-scattered electron (BSE) micrograph; b element distribution map). There are 300 × 225 quantitative points of the polished area with elements as shown on the map.

#### 3.3.1. Chemical Composition of Ye’Elimite

To obtain the chemical composition of ye’elimite, element contents were sorted and calculated from the ye’elimite phase area. The elemental composition of oxygen atoms was normalized to 16 for comparison. The results are presented in Table 4. The Al^3+^ is substituted by Fe^3+^ in all clinkers; however, the substitution rate was influenced by the CaO source. The content of Al^3+^ substituted by Fe^3+^ was much lower in clinker S00 than the other three. The maximum substitution rate of Fe^3+^ for Al^3+^ reached 17.34 wt.% in clinker S80, and the maximum proportion of Fe_2_O_3_ in the ye’elimite phase was 6.89 wt.%, expressed as C_4_A_2.71_F_0.29_$\bar{S}$. 

#### 3.3.2. Chemical Composition of Belite

The same method was used to study the chemical composition of the belite phase as was used for ye’elemite. The elemental composition of the belite phase area was calculated and oxygen atoms were normalized to four for comparison. The chemical composition of the belite phase is presented in Table 5. The result indicates that the amount of Fe_2_O_3_ solidified into belite was much lower than that in C_4_A_3_$\bar{S}$; the maximum value reached 1.86 wt.% in clinker S00. The main reason may be that Fe^3+^ is easier to substitute than Al^3+^ in C_4_A_3_$\bar{S}$ due to the same valence state and similar ionic radius rather than entering the lattice gap of C_2_S.

SO_3_ was also incorporated into C_2_S, and the incorporated amount increased with the increasing content of CaSO_4_ as the CaO sources in the raw mixture until 80%, and then decreased slightly when CaO was entirely sourced from CaSO_4_. The maximum incorporation content of SO_3_ reached 5.58 wt.% in clinker S80 but was 4.65 wt.% in clinker S100. It is worth noting that the crystal type of belite is closely related to the SO_3_ content incorporated into the C_2_S. Usually, belite exists as α’-C_2_S at 830–1470 °C and transforms to β-C_2_S at temperatures between 520 °C and 670 °C. Finally, it transforms to γ-C_2_S below 520 °C. However, the crystal transformation of C_2_S can slow down or even not occur under the influence of CaSO_4_. In addition, C_2_S obtained from Equations (5) and (6) exists as β-C_2_S but as α’-C_2_S from Equation (10) according to [15]. In clinker S00, the CaSO_4_ is exhausted after reacting with calcium aluminate or gehlenite to form ye’elimite that no more CaSO_4_ exist and the reactions of Equations (9) and (10) will not happen. Thus, the belite in clinker S00 exists as β-C_2_S. It is obvious that the increasing CaSO_4_ content induces more $4\mathrm{CaO}\cdot 2{\mathrm{SiO}}_{2}\cdot {\mathrm{CaSO}}_{4}$ formed as a transition phase and finally decomposed as α’-C_2_S [15]. Therefore, more α’-C_2_S is formed at the expense of β-C_2_S with the increase in CaSO_4_ as a CaO source. However, the decomposition of CaSO_4_ in S100 may be faster than in the others and CaSO_4_ rarely exists in the clinker. The reactions of Equations (9) and (10) are hampered, and the proportion of α’-C_2_S decreases with the increased CaSO_4_ in the raw mixture. 

#### 3.3.3. Chemical Composition of Ferrite Phase

The ferrite phase is a solid solution that mainly contains C_2_F, C_6_AF_2_, C_4_AF, and C_6_A_2_F. It is generally believed that iron is distributed in the CSA clinker in the form of C_4_AF but mainly exists as C_6_AF_2_ in the IR-CSA clinker. Iron distributed in the ferrite phase is presented in Figure 7 and the EPMA confirms its inhomogeneous distribution. Iron decreased from the center to the edge in all clinkers, which is consistent with the results of [14]. The ferrite phase is uniformly dispersed around the ye’elimite in clinker S00 and S50, but is concentrated in several areas as small clumps in S80 and S100. The content of the ferrite phase decreased with the increase in CaSO_4_ as the CaO source. To obtain the chemical composition of the ferrite phase, oxygen atoms were normalized to 15 for comparison. The results (presented in Table 6) indicate that the chemical composition in the ferrite phase is close to C_4_AF in S00 and S50 but close to C_6_AF_2_ in clinker S80 and S100. Although there is less iron in the form of the ferrite phase because of the incorporation into ye’elimite in clinkers S80 and S100 than in clinkers S00 and S50, CA is more likely to combine with CaSO_4_ Equation (3) than C_2_F Equation (8) or C_6_AF_2_ Equation (11) with the increasing CaSO_4_ content in the raw mixture. Thus, the ferrite phase in clinkers S80 and S100 has a chemical composition with lower aluminum but higher iron content than in clinkers S00 and S50.

### 3.4. Microstructural Characterization

Figure 8 shows the scanning electron micrograph of the section of the clinker sintered at 1300 ℃ for 45 min. All samples are magnified 5000 times. The hexagonal platy structure or quadrilateral columnar structure can be seen, and the particle size of C_4_A_3-*x*_F*_x_*$\bar{S}$ in S00 is about 10 μm, decreasing drastically in clinker S10 and then increasing gradually in S100. The iron-rich molten phase can be seen clearly around the ye’elimite in S00, which indicates a large amount of liquid ferrite phase during the process. However, the molten phase could hardly be observed in any other clinker with CaSO_4_ as the CaO source.

### 3.5. Compressive Strength of the IR-CSA Cement

The compressive strength of IR-CSA cement clinkers S00–S100 after 1 d, 3 d, and 28 d of curing is shown in Figure 9. With the increase in CaO derived from CaSO_4_, the 1 d and 3 d compressive strength exhibited slight strength advantages in cements S50 and S80, but disadvantages in S100. The 28d compressive strength was higher for cement S50–S100 than for S00–S20. To be specific, the cement in S50 showed the best early and late compressive strength. The main reasons may be as follows: (ⅰ) The anhydrate in cements S50 and S80 consumes more water for ye’elimite hydration than gypsum according to Equation (14); thus, the actual water/cement ratios of cements S50 and S80 were lower than other samples, affecting their mechanical properties; (ⅱ) The Al/Fe ratio of the ferrite phase of S50 (1:1.12) was higher than S80 (1:1.96) and S100 (1:2.12), which resulted in higher hydration activity of the ferrite phase and early compressive strength of the S50 than the S80 and S100. Furthermore, the C_4_A_3-*x*_F*_x_*$\bar{S}$ particle size of S50 was smaller than S80 and S100, which was beneficial for the hydration rate and early compressive strength; (ⅲ) The late compressive strength was related to the hydration of C_4_A_3-*x*_F*_x_*$\bar{S}$ and C_2_S. With the increase in CaSO_4_ as a CaO source, the content of C_4_A_3-*x*_F*_x_*$\bar{S}$ in clinkers S50–S100 was higher than in S00–S20. In addition, the C_2_S hydration activity in S50 and S80 is higher than in the other cements because of the crystal type (the crystal type of α’-C_2_S exhibits higher hydration activity than β-C_2_S at room temperature). 

## 4. Conclusions

The effect of CaO sources, from either CaCO_3_ or CaSO_4_, on phase formation and mineral composition of iron-rich clinker was investigated by varying their proportions in raw materials. Compared with CaCO_3_, CaO derived from CaSO_4_ was more conducive for Al_2_O_3_ and Fe_2_O_3_ to form C_4_A_3-*x*_F_x_$\bar{S}$ rather than the ferrite phase which resulted in an increased content of ye’elimite but a decreased ferrite phase. More c-C_4_A_3-*x*_F*_x_*$\bar{S}$ formed at the expense of o-C_4_A_3-*x*_F*_x_*$\bar{S}$ in the clinker because of the incorporation of Fe_2_O_3_. In addition, the belite phase was more inclined to exhibit as α’-C_2_S instead of β-C_2_S when taking CaSO_4_ as a CaO source, which is conducive to the hydration activity of the clinker. Finally, the crystal grain sizes decreased dramatically and then increased gradually with the increase in the proportion of CaO derived from CaSO_4_. In view of the influence of the mineral composition, crystal structure and crystal grain sizes, cement with optimal mechanical properties was obtained with the proportion of 1:1 of CaCO_3_ and CaSO_4_ as a CaO source.

Under the specified conditions, the substitution of Fe^3+^ for Al^3+^ reached a maximum value of 17.34 wt.%, and the maximum proportion of Fe_2_O_3_ in the ye’elimite phase was 6.89 wt.%, expressed as C_4_A_2.71_F_0.29_$\bar{S}$. However, the incorporation amount of Fe_2_O_3_ into C_2_S was no more than 1.86 wt.% and showed an irregular change trend with CaO sources. The chemical formula of the ferrite phase was calculated and found to be similar to C_4_AF when the amount of CaO derived from CaSO_4_ was less than 50% but similar to C_6_AF_2_ when the amount of CaO derived from CaSO_4_ was more than 80%, which also indicates that Al_2_O_3_ is more inclined to be involved in C_4_A_3-*x*_F*_x_*$\bar{S}$ with the increase in CaSO_4_ as a CaO source.

The findings provide a possible method to optimize the mineral composition of IR-CSA clinker by adjusting the content of CaSO_4_ as a CaO source and to produce high-performance IR-CSA cement at a low cost through cooperative utilization of waste gypsum and iron-bearing industrial solid wastes.

## Figures and Tables

**Figure 1 materials-16-00643-f001:**
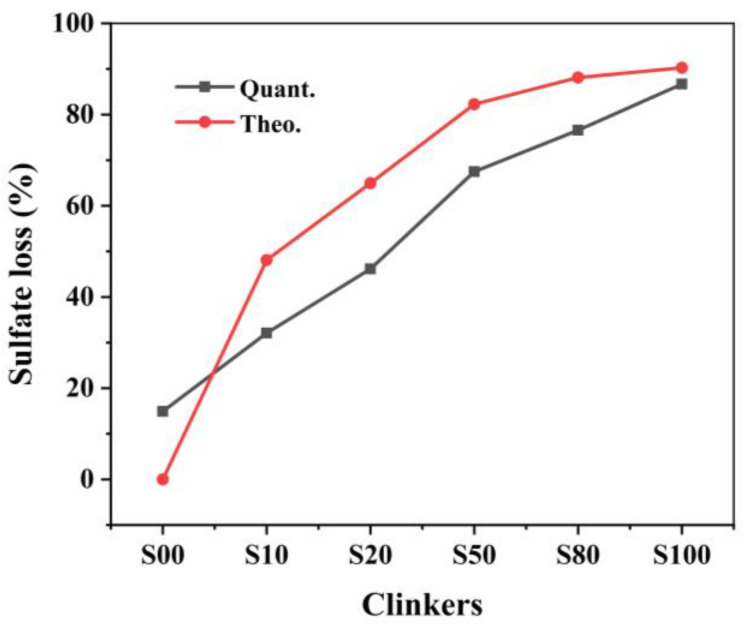
Quantitative and theoretical sulfate loss in the clinker with different CaO sources.

**Figure 2 materials-16-00643-f002:**
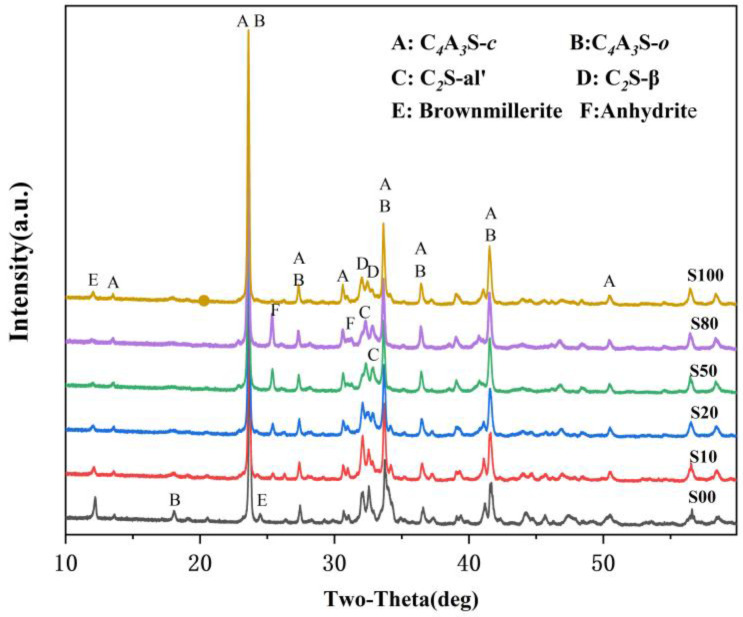
XRD patterns of clinkers S00−S100.

**Figure 3 materials-16-00643-f003:**
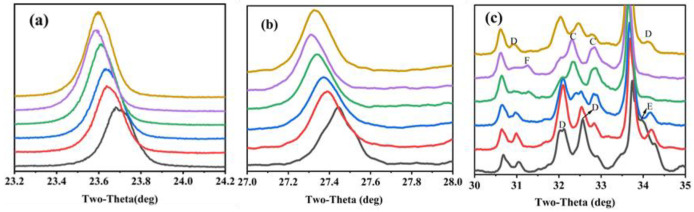
Zoomed section of characteristic peaks migrate with variation proportion of CaSO_4_ as a CaO source (**a**) ye’elimite at about 23.7°, (**b**) ye’elimite at about 27.4° and (**c**) belite-al’(C), belite-beta (D), brownmillerite (E) and anhydrate (F).

**Figure 4 materials-16-00643-f004:**
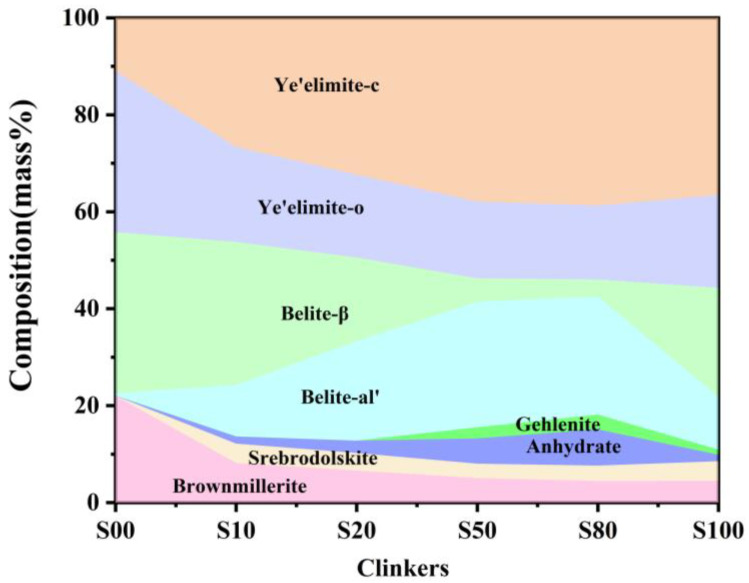
Phase evolution of clinkers S00−S100.

**Figure 5 materials-16-00643-f005:**
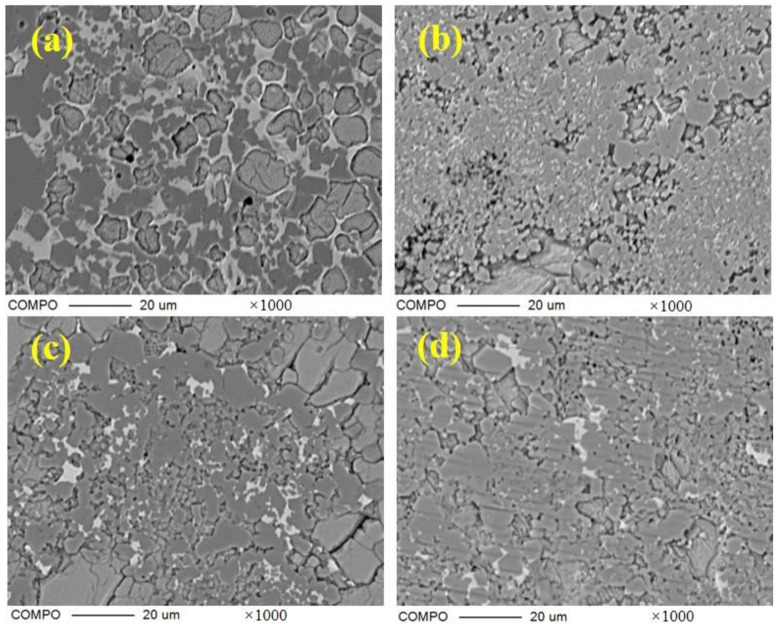
BSE micrograph of polished section of samples with 1000 times magnification; (**a**) S00; (**b**) S50; (**c**) S80; (**d**) S100.

**Figure 6 materials-16-00643-f006:**
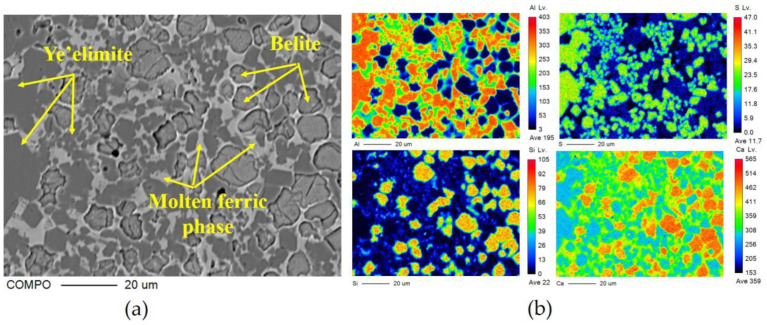
EPMA results of clinker S00: (**a**) BSE micrograph of polished section; (**b**) distribution of all measured elements (Al, S, Si and Ca).

**Figure 7 materials-16-00643-f007:**
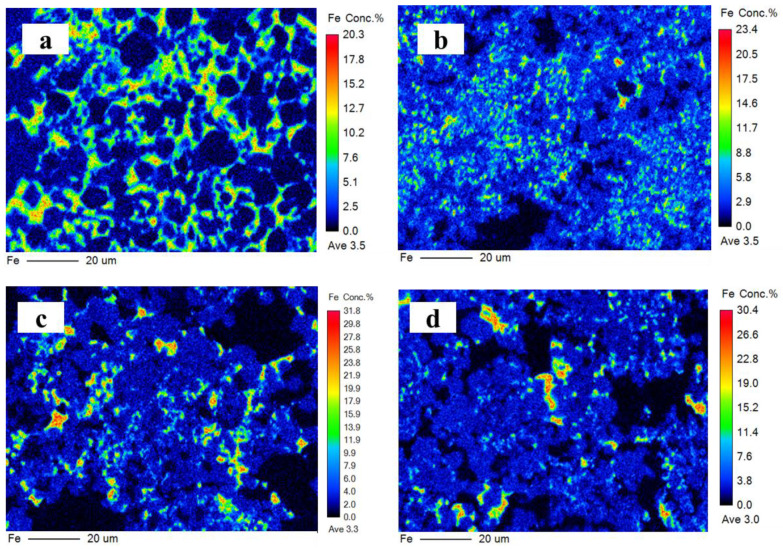
Iron distribution on polished surface of clinker S00 (**a**), S50 (**b**), S80 (**c**), and S100 (**d**).

**Figure 8 materials-16-00643-f008:**
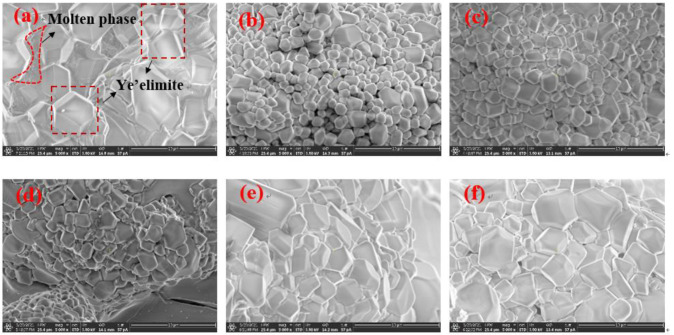
Scanning electron micrograph of section of clinkers S00 (**a**), S10 (**b**), S20 (**c**), S50 (**d**), S80 (**e**) and S100 (**f**).

**Figure 9 materials-16-00643-f009:**
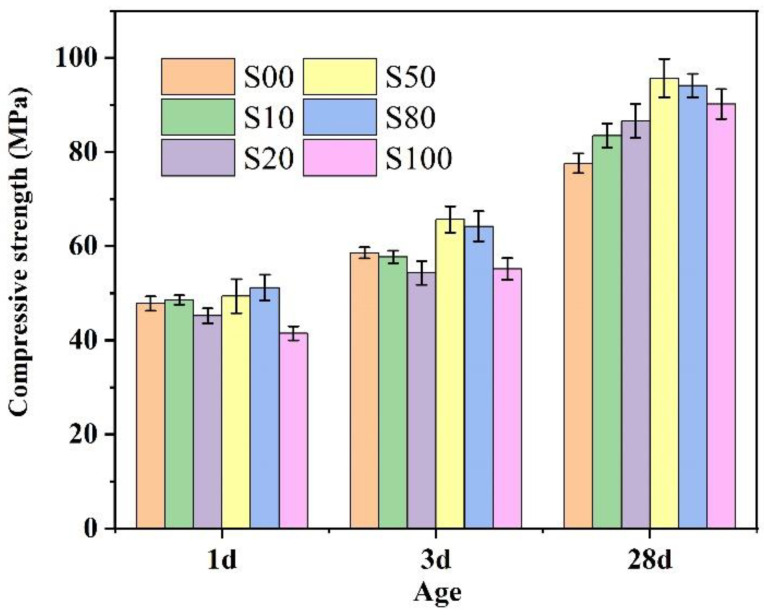
Compressive strength of cements S00–S100.

**Table 1 materials-16-00643-t001:** Target mineral composition of the clinker and chemical reagent proportions in the raw materials to prepare 100 g clinker.

Sample	Targeted Minerals			Raw Material Proportion				
	$C_{4}A_{3}\bar{S}$	C_2_S	C_4_AF	CaCO_3_	SiO_2_	Al_2_O_3_	Fe_2_O_3_	CaSO_4_·2H_2_O
S00	50	30	20	55.69	7.67	21.47	4.83	10.34
S10	50	30	20	48.19	7.37	20.64	4.64	19.15
S20	50	30	20	41.25	7.10	19.88	4.47	27.31
S50	50	30	20	23.20	6.39	17.89	4.02	48.51
S80	50	30	20	8.43	5.81	16.26	3.65	65.85
S100	50	30	20	0	5.47	15.33	3.44	75.76

**Table 2 materials-16-00643-t002:** Crystallographic structures and ICSD codes of phases involved in the Rietveld refinement.

Scientific Names	Crystal System	Space Group	Abbrev. of Formula	Chemical Formula	ICSD
Ye’elimite-*c*	cubic	I-43m	C_4_A_3_$\bar{S}$-c	Ca_4_Al_6_SO_16_	9560
Ye’elimite-*o*	orthorhombic	Pcc2	C_4_A_3_$\bar{S}$-o	Ca_4_Al_6_SO_16_	80,361
Belite-beta	Monoclinic	P21/*n*	C_2_S-beta	Ca_2_SiO_4_	79,550
Belite-al’	orthorhombic	Pnma	C_2_S-al’	Ca_2_SiO_4_	81,097
Srebrodolskite	orthorhombic	Pnma	C_2_F	Ca_2_Fe_2_O_5_	15,059
Brownmillerite	orthorhombic	Ibm2	C_4_AF	Ca_4_Al_2_Fe_2_O_10_	9197
Gehlenite	Tetragonal	*p*-421m	C_2_AS	Ca_2_Al_2_SiO_7_	87,144
Anhydrite	orthorhombic	Amma	$C\bar{S}$	CaSO_4_	16,382

**Table 3 materials-16-00643-t003:** CaO and SO_3_ content of clinkers and their differences between quantitative and theoretical values.

$\mathrm{CaO}/{\mathrm{SO}}_{3}(\Delta )$ Content	Theo.	S00	S10	S20	S50	S80	S100
SO_3_	6.56	5.58	8.58	10.08	12.01	12.93	8.96
$\Delta \left({\mathrm{SO}}_{3}\right)=S\left({\mathrm{SO}}_{3}\right)-\;\mathrm{Theo}.\left({\mathrm{SO}}_{3}\right)$	–	−0.98	2.02	3.52	5.46	6.37	2.40
CaO	42.53	43.22	41.12	40.07	38.71	38.07	40.85
$\Delta \left(\mathrm{CaO}\right)=S\left(\mathrm{CaO}\right)-\;\mathrm{Theo}.\left(\mathrm{CaO}\right)$	–	0.69	−1.41	−2.46	−3.82	−4.46	−1.68

**Table 4 materials-16-00643-t004:** Mineralogical composition of ye’elimite phase of samples S00, S50, S80 and S100.

Sample	Mineralogical Composition	Fe/(Al + Fe)(wt.%)	Fe_2_O_3_ in Ye’elimite Phase(wt.%)
S00	Ca_4.0_Al_5.82_ Fe_0.17_Si_0.07_S_0.96_O_16_	5.71	2.19
S50	Ca_3.94_Al_5.17_ Fe_0.45_Si_0.13_S_1.08_O_16_	15.29	4.91
S80	Ca_3.93_Al_5.24_Fe_0.53_Si_0.13_S_1.12_O_16_	17.34	6.89
S100	Ca_4.06_Al_5.09_Fe_0.46_Si_0.14_S_1.12_O_16_	15.79	6.03

**Table 5 materials-16-00643-t005:** Mineralogical composition of belite phase of samples S00, S50, S80 and S100.

Sample	Mineralogical Composition	Fe_2_O_3_ in Belite Phase (wt.%)	SO_3_ in Belite Phase (wt.%)
S00	Ca_1.97_Al_0.08_ Fe_0.04_Si_0.85_S_0.05_O_4_	1.86	2.32
S50	Ca_1.96_Al_0.01_ Fe_0.02_Si_0.81_S_0.12_O_4_	0.93	5.56
S80	Ca_1.92_Al_0.01_Fe_0.01_Si_0.91_S_0.11_O_4_	0.47	5.58
S100	Ca_1.87_Al_0.05_Fe_0.02_Si_0.84_S_0.14_O_4_	0.93	4.65

**Table 6 materials-16-00643-t006:** Chemical composition of ferrite phase of samples S00, S50, S80, and S100.

Sample	Chemical Composition	Fe/Al	Abbreviations
S00	Ca_6.31_Al_2.98_ Fe_2.82_O_15_	0.95	C_4.23_AF_0.95_
S50	Ca_6.32_Al_2.68_Fe_2.98_O_15_	1.12	C_4.71_AF_1.12_
S80	Ca_6.46_Al_1.92_Fe_3.77_O_15_	1.96	C_6.73_AF_1.96_
S100	Ca_6.47_Al_1.82_Fe_3.87_O_15_	2.12	C_7.11_AF_2.12_

## Data Availability

The data presented in this study are available on request from the corresponding author.

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
