# Peer review of "Effect of CaO Sourced from CaCO3 or CaSO4 on Phase Formation and Mineral Composition of Iron-Rich Calcium Sulfoaluminate Clinker"

_materials, 2023, doi:10.3390/ma16020643_

Round 1

Reviewer 1 Report

Very well written and conducted research project. Just a few minor clarifications.

1. Please mention the manufacturer details of the Muffle furnace.

2. Kindly provide details of the sample mould and how the samples were retrieved.

3. Mention the magnification of the EPMA and SEM micrographs.

4. What ISO specification was followed for the compressive strength evaluation and why the 28 day CS evaluation.

Reviewer 2 Report

Dear Authors,

the article deals with the effect of CaO sources from CaCO3 or CaSO4 on iron-rich clinker when changing the proportions in the raw materials.  Study results provide theoretical guidance for the production of high-performance IR-CSA cement through the synergistic use of waste gypsum and iron-containing ISWs. This may have the potential to reduce pollutant emissions and clinker burning temperatures.

From my point of view, this is an interesting article and has a very high scientific value. Compared to other published materials, the topic is original and the issue is very important, especially for the protection of the natural environment.

The experimental program and the results obtained were adequately described. The literature is not very extensive, but it is correctly cited. The article requires a minor editorial correction and a few explanations, which I have attached and marked in yellow. Conclusions should be more transparent and bulleted.

After completing the suggestions, I recommend the article for publication.

Kind regards.

Reviewer 3 Report

1.      This study evaluates the effect of variation in CaO sources on phase formation and engineering properties of calcium sulfoaluminate clinkers. The topic of the study is interesting and the manuscript has significant contributions provided by solid testing programs. Nevertheless, the English language of the study is extremely poor and cannot be accepted in its current form. Authors should revise the entire manuscript to improve the English and editorial quality of the manuscript before considering it for publication.

2.      The English language needs to be polished due to several minor typos and grammatical errors. Below are some examples:

·        Page 1 Line 20: It should be “was influenced by”.

·        Page 1 Line 22: What does it mean: “…is less than 50 at.% but as…”. If authors mean atomic ratio, they need to define it since it is not a common acronym.

·        Page 1 Line 24: It should be “… by synergistic use/utilization of…

·        Page 2 Line 33: It should be “CSA cement

·        Page 2 Line 34: It should be “lower than that of PC clinker

·        Page 2 Line 36: It should be precast, not pre-cast.

·        Page 2 Line 38: It should be “…CAS cement will be a …

·        There should be a space between units and numbers: 100 g not 100g.

·        There should be a space between the last word of the sentence and the given reference: raw mixture [17]. Not raw mixture[17].

As it is evident, there are a huge number of English faults in the manuscript which need a major revision preferably by a native or expert in English.

3.      What is FR-clinker in Line 130 on Page 4? On page one you mentioned IR-CSA. Are they the same? Please make sure to check the entire manuscript for a careful introduction of acronyms when you use them for the first time.

4.      The quality of the Figures is very low and needs to be improved.

5.      Delete the legend in Figure 6a and indicate the phases on the image using arrows.

6.      Please insert a scale bar in Figs. 5 and 6, similar to Fig. 8.

7.      The abstract and conclusions need to be revised to better reflect the objectives and findings of the study.

8.      The poor English language makes it difficult to understand several parts of the discussions. Therefore, the technical content of the manuscript may need further evaluation after the English revision.
